# MITOL deletion in the brain impairs mitochondrial structure and ER tethering leading to oxidative stress

Shun Nagashima[1,*], Keisuke Takeda[1,*], Nobuhiko Ohno[2], Satoshi Ishido[3], Motohide Aoki[4], Yurika Saitoh[5], Takumi Takada[1], Takeshi Tokuyama[1], Ayumu Sugiura[1], Toshifumi Fukuda[1], Nobuko Matsushita[1], Ryoko Inatome[1], Shigeru Yanagi[1]

**Mitochondrial abnormalities are associated with developmental disorders, although a causal relationship remains largely unknown. Here, we report that increased oxidative stress in neurons by deletion of mitochondrial ubiquitin ligase MITOL causes a potential neuroinflammation including aberrant astrogliosis and microglial activation, indicating that mitochondrial abnormalities might confer a risk for inflammatory diseases in brain such as psychiatric disorders. A role of MITOL in both mitochondrial dynamics and ER-mitochondria tethering prompted us to characterize three-dimensional structures of mitochondria in vivo. In MITOL-deficient neurons, we observed a significant reduction in the ER-mitochondria contact sites, which might lead to perturbation of phospholipids transfer, consequently reduce cardiolipin biogenesis. We also found that branched large mitochondria disappeared by deletion of MITOL. These morphological abnormalities of mitochondria resulted in enhanced oxidative stress in brain, which led to astrogliosis and microglial activation partly causing abnormal behavior. In conclusion, the reduced ER-mitochondria tethering and excessive mitochondrial fission may trigger neuroinflammation through oxidative stress.**

## Introduction

Precise neuronal network formation during brain development assures not only normal ontogeny but also higher brain functions including thinking, behaviors and memory. However, abnormalities in neurogenesis, neuronal cell migration, neuroinflammation and synapse formation lead to aberrant neuronal network, causing developmental disorders such as autism spectrum disorder (ASD) (Reiner et al, 2016). Developing neurons require high mitochondrial energy production to construct complicated neural circuits through proper neuronal cell migration and dynamic regulation of axon guidance called "scrap and build" (Lathrop & Steketee, 2013; Lin & Sheng, 2015). Conversely, high demand of mitochondrial respiratory activity is accompanied by a risk of oxidative stress due to increased electron leak from mitochondrial respiratory chain under physiological and pathological changes impaired mitochondrial homeostasis. Thus, high quality mitochondria are required for correct brain development and functions thereafter. It has been reported that mitochondrial dysfunction is associated with developmental disorders (Frye & Rossignol, 2011; Rossignol & Frye, 2012), although a causal relationship is unclear at present. It is therefore possible that mitochondrial abnormalities are involved in either the pathology or the potential risk of developmental disorders.

Mitochondria dynamics repeating fusion and fission is a key machinery to maintain mitochondrial homeostasis. In addition, morphological changes of mitochondria are important to release of cytochrome c from mitochondria, inducing apoptosis. Drp1 is an essential modulator of mitochondrial fission. Recent studies have also identified some proteins to function as Drp1 receptors, named Mff and MiD49/51. The unique regions of the ER connected with mitochondria is known as the ER-mitochondria contact sites (Franke & Kartenbeck, 1971; Morre et al, 1971; Vance, 1990). Accumulating evidence suggest that the proximal junction between the ER and mitochondria plays multiple, important cellular functions not only in the efficient transfer of $Ca^{2+}$ from the ER to the mitochondria and lipid metabolism but also the formation of the autophagic isolation membrane, cell death signaling and other processes (Vance, 1990, 2014; Simmen et al, 2005; Szabadkai et al, 2006; Hayashi & Su, 2007; Kornmann et al, 2009; Horner et al, 2011; Zampese et al, 2011; Zhou et al, 2011; Rowland & Voeltz, 2012; Hamasaki et al, 2013; Schon & Area-Gomez, 2013; Prudent et al, 2015). In yeast, the ER-mitochondrial encounter structure (ERMES), a tethering complex that bridges the ER and mitochondria, has been clarified to be involved in phospholipid transport (Kornmann et al,

[1]Laboratory of Molecular Biochemistry, School of Life Sciences, Tokyo University of Pharmacy and Life Sciences, Hachioji, Tokyo, Japan    [2]Department of Anatomy, Division of Histology and Cell Biology, School of Medicine, Jichi Medical University, Shimotsuke, Japan    [3]Department of Microbiology, Hyogo College of Medicine, Nishinomiya, Japan    [4]Laboratory of Bioanalytical and Environmental Chemistry, School of Life Sciences, Tokyo University of Pharmacy and Life Sciences, Hachioji, Tokyo, Japan    [5]Department of Tokyo Physical Therapy, Faculty of Medical Science, Teikyo University of Science, Adachi-ku, Tokyo, Japan

Correspondence: syanagi@ls.toyaku.ac.jp
*Shun Nagashima and Keisuke Takeda contributed equally to this work.

2009). However, in mammals, the in vivo structure and function of ER-mitochondria contact sites are largely unknown.

Previously, we have identified the mitochondrial ubiquitin ligase (MITOL, also known as MARCH5); an integral mitochondrial outer membrane protein with four membrane-spanning segments that is a member of the membrane-associated RING-CH E3 ubiquitin ligase (MARCH) family (Nakamura et al, 2006; Yonashiro et al, 2006; Nagashima et al, 2014). MITOL controls mitochondrial dynamics by regulating mitochondrial fission factors, such as Drp1 and Mid49 (Yonashiro et al, 2006; Karbowski et al, 2007; Xu et al, 2016). Furthermore, recent studies have suggested that MITOL has several functions including maintenance of embryonic stem cells stemness, cellular senescence, cell survival, and immune responses via regulation of mitochondrial antiviral signaling protein (Park et al, 2010; Park et al, 2014; Gu et al, 2015; Yoo et al, 2015). Mitofusin2 (Mfn2) has been reported to induce ER-mitochondria tethering by Mfn2 oligomer formation, although a role of Mfn2 in the proximal junction between the ER and mitochondria remains controversial in cellular experiments (de Brito & Scorrano, 2008; Chen et al, 2012; Cosson et al, 2012; Sebastian et al, 2012; Schneeberger et al, 2013; Filadi et al, 2015; Wang et al, 2015; Naon et al, 2016) Previously, we have demonstrated that MITOL mediates lysine-63-linked poly-ubiquitin chain addition to Mfn2 and induces ER tethering to mitochondria through Mfn2 oligomerization. Thus, MITOL regulates ER tethering to mitochondria by activating Mfn2 (Sugiura et al, 2013).

In the present study, we analyzed nerve-specific MITOL KO mice for the first time, and report that MITOL KO in brain reduces the average volume of mitochondria and impairs the formation of ER-mitochondria contact sites. Moreover, MITOL KO confers a risk of developmental disorders by oxidative stress-induced neuro-inflammation. We also suggest a relationship between morphological abnormalities of mitochondria and developmental disorder.

# Results

### Loss of large and branched mitochondria in MITOL KO neurons

MITOL is a key regulator for mitochondrial dynamics. We and other group reported that down-regulation of MITOL in cultured cells induced mitochondrial fission through increased Drp1 and Mid49 (Yonashiro et al, 2006; Xu et al, 2016). To elucidate MITOL function in mitochondrial dynamics in vivo, we generated cerebral cortex-, hippocampus-, and olfactory bulb-specific MITOL KO mice (eKO) by crossing MITOL flox/flox mice (wild type: WT) with Emx1-Cre mice (Fig S1A). The specific deletion of the *march5/mitol* gene and protein expression in the cerebral cortex of the mutant mouse was confirmed by both immunoblot and genomic PCR (Fig S1B and C). The eKO are viable and fertile. No significant difference between WT and eKO was observed in both body size and brain size (Fig S1D and E). In order to understand precise morphology of mitochondria in MITOL KO hippocampal neurons, we performed three-dimensional (3D) reconstructions from serial EM images of mitochondria using Serial block-face scanning EM (SBF-SEM) (Fig S1F and G). The 3D structures of over 300 mitochondria in MITOL KO neurons were reconstructed and compared with those from WT neurons;

characteristic WT and MITOL KO neuron mitochondrial morphologies are shown in Fig 1A. Importantly, small and simple structured mitochondria were dominant, while large branched mitochondria were scarcely detected in MITOL KO neurons. Statistical analyses demonstrated that mitochondrial volume and branch number were drastically reduced in the cell body of MITOL KO neurons (Fig 1B and C). No obvious change was observed in the correlation between mitochondrial volume and sphericity, indicating that prominent mitochondrial swelling is absent in MITOL KO neurons (Figs 1D and S1H). We and others have previously reported that MITOL ubiquitinates mitochondrial fission proteins and promotes their proteasomal degradation. Consistently, immunoblot analysis showed that Drp1 was slightly increased in MITOL KO crude mitochondrial fraction (Fig 1E and F). It has been widely accepted that phosphorylation of Drp1 at Ser616 triggers mitochondrial fission through its recruitment to mitochondria. The phosphorylated Drp1 at Ser616 was accumulated in MITOL KO brain (Fig S1I), suggesting that increased Drp1 by MITOL KO leads to mitochondrial fission. In addition, we compared Drp1 receptors Mid49 and Mff. Consistent with a previous report using cultured cells (Xu et al, 2016), Mid49 was accumulated in MITOL KO brain. On the other hand, FundC1, which was reported to be a substrate for MITOL (Chen et al, 2017), was not altered. We did not detect enhancement of autophagy and mitophagy by SBF-SEM analysis. Therefore, the increase of small mitochondria in MITOL KO neurons may be due to accumulated mitochondrial fission proteins.

### Reduced ER-mitochondria contact sites in MITOL KO neurons

It has been shown that ER-mitochondria contacts are induced by forming Mfn2/Mfn2 or Mfn2/Mfn1 oligomers (de Brito & Scorrano, 2008). Previously, we have demonstrated that MITOL ubiquitinates mitochondrial Mfn2 and promotes Mfn2 oligomer formation, leading to ER-mitochondria tethering in cultured cells. Initially, we examined the Mfn2 complex in this mutant mouse line. Blue-Native poly-acrylamide gel electrophoresis (BN-PAGE) analysis and sucrose density gradient centrifugation assays demonstrated that Mfn2 oligomerization was partially impaired in the MITOL KO cerebral cortex (Fig S2A and B); supporting our previous report that MITOL is required for Mfn2 activation. We next examined whether MITOL deletion alters the state of ER-mitochondria contact sites in vivo by SBF-SEM analysis. To identify proximal junction between the ER and mitochondria, the mitochondria and ER in each EM image were outlined in yellow and green, respectively, and ER membranes-attached mitochondria were labeled as ER-mitochondria contact sites in red (Fig S3A–F). Representative reconstructions of a mitochondrion (yellow), mitochondrion-attached partial ER (other colors), and ER-mitochondria contact sites (red) are shown in Fig S3G and H and Video 1. Interestingly, this 3D-analysis revealed several morphologically distinct types of ER-mitochondria contacts such as large or small one. Distinct size of ER-mitochondria contact sites might display distinct functions. The representative data were shown in Fig 2A. Statistical analyses of SBF-SEM data revealed a drastic reduction in both total area and the number of ER-mitochondria contact sites in MITOL KO mitochondria in the cell body (Fig 2B and C). No significant difference in each area of ER-mitochondria contact sites was observed between

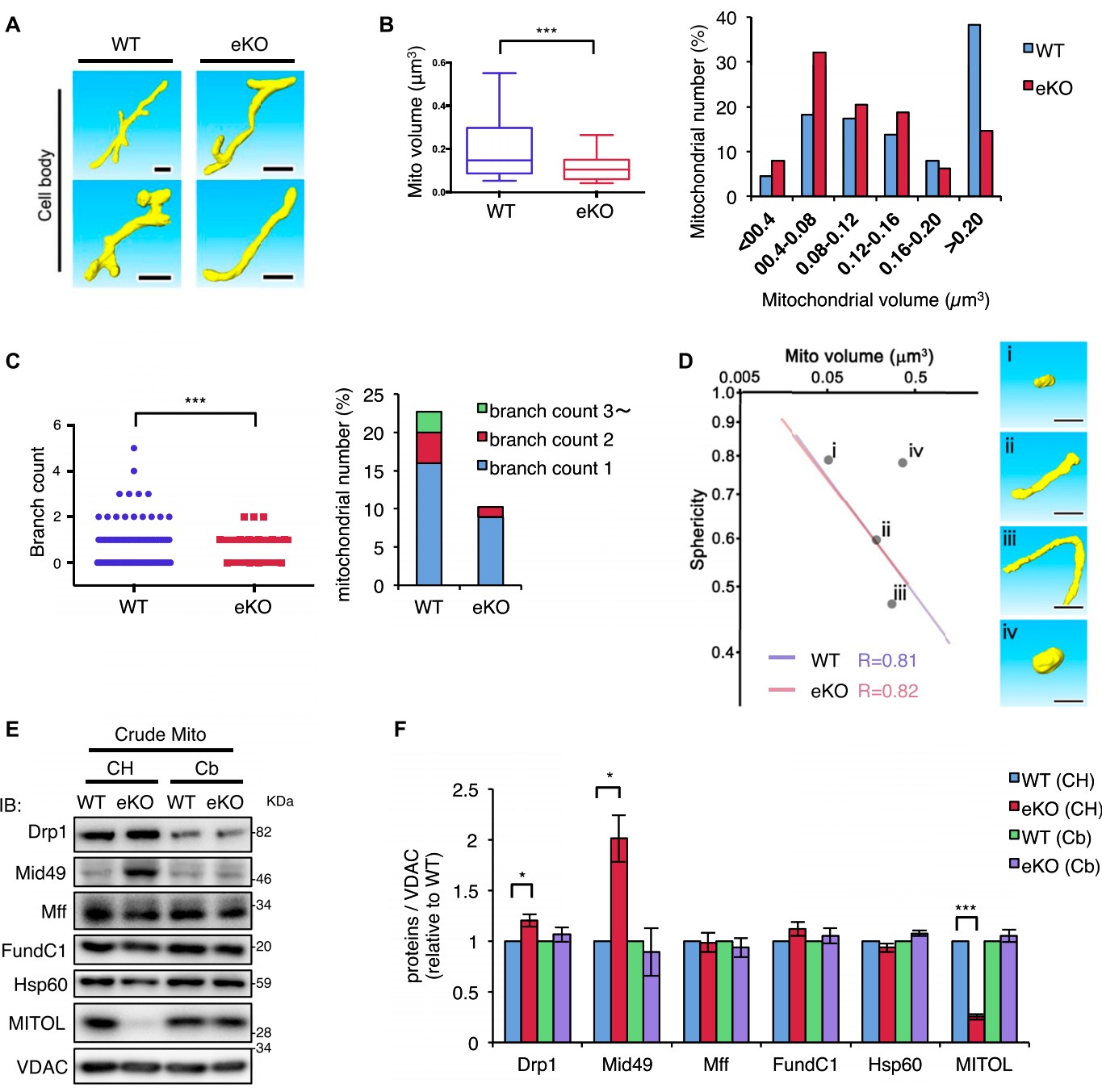

**Figure 1.   SBF-SEM analysis unveiled loss of large and branched mitochondria in MITOL KO neuron.**
**(A)** Representative images of mitochondrial morphology in WT and MITOL KO hippocampal neurons by SBF-SEM analysis. Scale bars represent 1 $\mu$m. **(B, C)** Statistical analyzes indicate reduced mitochondrial volume (B) and branch number (C) in the cell body of hippocampal neurons of WT and eKO. **(B)** Left scatter dot plot indicates mitochondrial volume. Horizontal bar, median; box limits, 25th and 75th percentiles; whiskers, 10th and 90th percentiles. Right bar graph indicates the distribsion of mitochondrial volume. **(C)** Left scatter dot plot indicates the number of mitochondria with indicated mitochondrial branch count. Right bar graph indicates the percentage of mitochondria with indicated mitochondrial branch count. (n = 225 for mitochondria in cell body; ***$P < 0.005$, U test). **(D)** No change in correlation between mitochondrial volume and sphericity. The volume and sphericity of representative mitochondria (right panels; i–iv) is plotted in the scatter plot (gray dots in the left panel, i–iv). (n = 225 for mitochondria in cell body). **(E, F)** Comparison of proteins involved in mitochondrial fission in mitochondrial fraction. Mitochondria isolated from combination of cerebral cortex and hippocampus of WT or eKO were analyzed by IB with the indicated antibodies. (n = 6–9; *$P < 0.05$, ***$P < 0.005$, t test). Error bars indicate SEM. CH, Cerebral cortex and hippocampus; Cb, Cerebellum.

WT and MITOL KO mitochondria in the cell body (Fig 2D). To rule out the possibility that ER-mitochondria contact sites were reduced by loss of mitochondrial volume in MITOL KO neurons, we assessed the ratio of ER-mitochondria contact area per mitochondrial surface area and found a reduced ratio of ER-mitochondria contact area per mitochondrial surface in MITOL KO mitochondria (Fig 2E). Moreover,

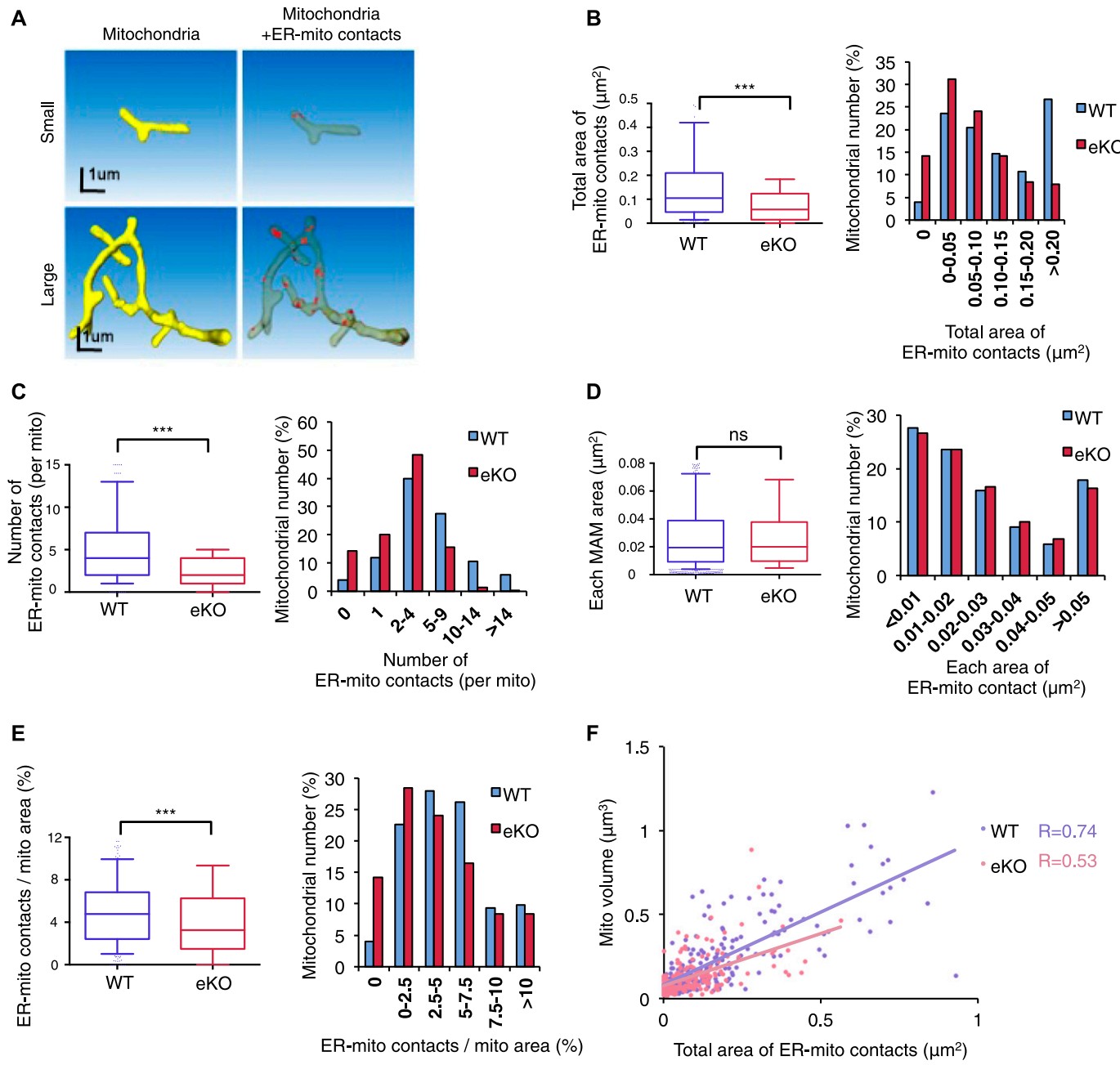

**Figure 2. Reduced ER tethering to mitochondria in MITOL KO neurons.**
**(A)** 3D reconstruction of the mitochondrion (yellow) with ER-mitochondria contact sites (red) in WT hippocampal neurons by SBF-SEM analysis. **(B–E)** Reduced ER-mitochondria contact sites in eKO. The graphs indicate total area and number of ER-mitochondria contact sites per mitochondrion (B, C), and each ER-mitochondria contact area (D) and total area of ER-mitochondria contact sites per mitochondrial surface area (E). Data are shown as 10–90% bocplots with the 25th, 50th, and 75th percentiles as the lower, middle, and upper boundaries of the box in the left graphs (n = 225 for mitochondria in cell body; **$P < 0.01$, ***$P < 0.005$, U test). Right bar graphs indicate the distribusion. **(F)** Disrupted correlation between total area of ER-mitochondria contact sites and mitochondrial size in neuronal cell bodies of eKO mice. Spearman's rank correlation coefficient (R) represents the strength of a relationship between total area of ER-mitochondria contact sites and mitochondrial volume. (n = 225).

the correlation between the area of ER-mitochondria contact sites and mitochondrial size was disrupted in MITOL KO neurons (Fig 2F), indicating that the frequency of ER-mitochondria contact is reduced in MITOL KO mitochondria. Taken together, these findings indicate that MITOL regulates ER-mitochondria tethering in vivo.

## Abnormal mitochondrial contents in MITOL KO neurons

We next examined internal structural alterations of mitochondria in MITOL KO neurons. SBF-SEM analysis demonstrated severely disrupted structures of mitochondria including numerous dilated,

swollen cristae and empty matrices (Fig 3A). SBF-SEM analysis was able to detect the abnormalities of mitochondrial internal structures in the different sections of the same mitochondrion. The bar graph showed the number of abnormal mitochondrial sections in each mitochondrion (Fig 3B). The abnormalities were observed in 50% of mitochondria in MITOL KO neurons in contrast to 30% of mitochondria in WT neurons. Fig 3C illustrates the cumulative frequency of abnormal mitochondrial morphology per mitochondrial unit volume. In MITOL KO neurons, morphological abnormality was observed with higher frequency in small mitochondria than in large mitochondria. CL is the major phospholipid in mitochondrial inner membrane and has been shown to be required for maintenance of mitochondrial morphology; in particular, the cristae structure (Saric et al, 2015). In order to measure the amount of CL, the mitochondrial phospholipids in WT and MITOL KO brains were compared by thin-layer chromatography (TLC) (Figs 3D and S4A). Interestingly, CL was reduced in mitochondria isolated from MITOL KO brains. In contrast, there were no significant differences observed for the other phospholipids including phosphatidylethanolamine, phosphatidylserine and phosphatidylcholine. To further analyze CL levels, we performed liquid chromatography-mass spectrometry (LC/MS) analysis. Our results revealed a significant

reduction in nearly all CL species that are highly expressed in the brain (Figs 3E and S4B). Taken together, MITOL deficiency causes mitochondrial disruption, such as swollen cristae, partly through reduced CL.

## Enhanced astrogliosis and microglial activation in MITOL-KO brain

A recent study has indicated that MITOL KO in cultured cells induces mitochondrial fragmentation, involving in apoptosis induction in a stress dependent manner rather than mitochondrial dysfunction. Actually, the previous study has showed the same level of the cellular oxygen consumption rate and extracellular acidification rate between WT and MARCH5/MITOL KO cells (Xu et al, 2016) suggesting that MITOL KO does not alter mitochondrial respiratory activity in the cultured cell. In contrast, MITOL KO in mice displayed a mild reduction of CL and impairment of mitochondrial internal structure, which had been reported to highly correlate to mitochondrial respiratory activity. Thus, we examined the mitochondrial respiratory activity in the brain. MITOL KO in brain induced a slight decrease of the formation and activity of mitochondrial respiratory complexes and supercomplexes

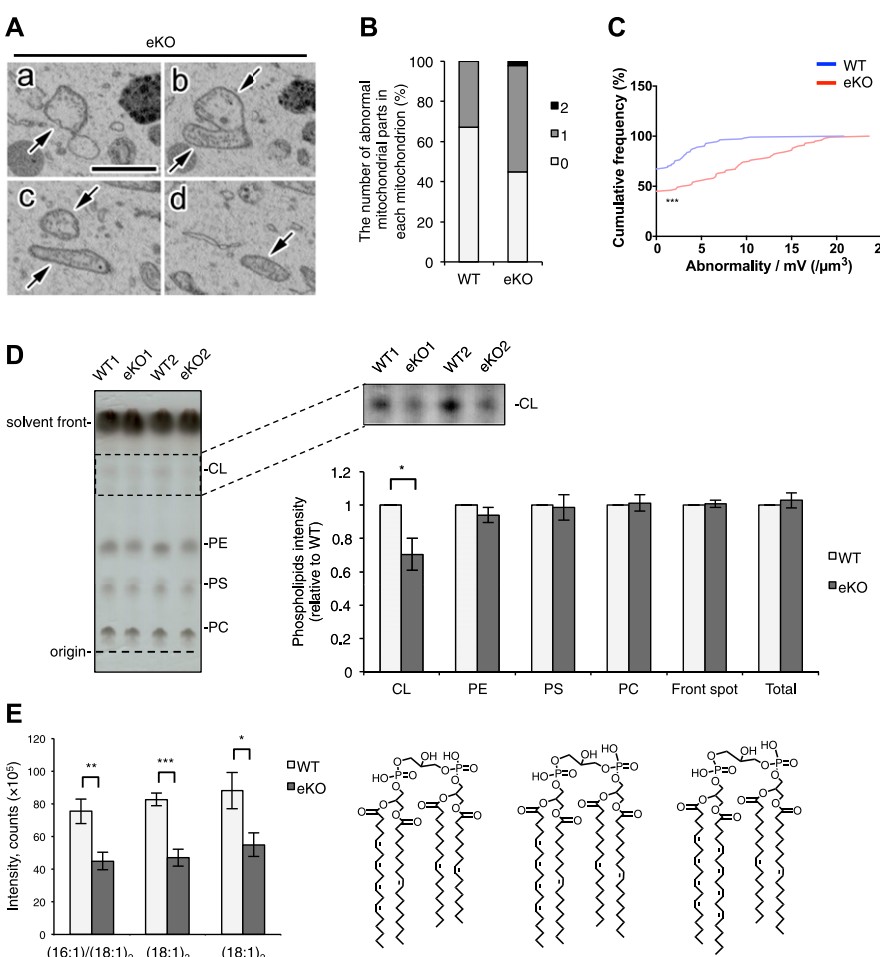

**Figure 3. Reduced cardiolipin and disrupted cristae in MITOL KO neurons.**
**(A–C)** Abnormal structures ("abnormality") of mitochondria (arrows) in the SBF-SEM analysis of hippocampal neurons of 12-wk-old eKO (A). **(A–D)** The serial images are shown in (a–d). The connection between the abnormally swollen part and the normally appearing region is clearly observed in the serial images (a–d, arrows). Scale bar represents 2 μm. The bar graph showed the number of abnormal mitochondrial parts in each mitochondrion (B). Cumulative frequency of abnormal mitochondria per unit volume (C). More mitochondria have higher frequency of abnormality per unit mitochondrial volume in eKO. (n = 110 for mitochondria in cell body of WT and n = 100 for mitochondria in cell body of eKO; ***$P$ < 0.001, U test). All error bars indicate SEM. **(D)** TLC separation of mitochondrial phospholipids isolated from combination of cerebral cortex and hippocampus. The right upper panel is processed from left panel to analyze CL. The right lower graph shows quantifications of each phospholipid in equal amount of mitochondrial fraction isolated from WT and eKO. (n = 5; *$P$ < 0.05, $t$ test). **(E)** Assessment of representative molecular species of CL in the same sample as above by LC/MS. (n = 5; *$P$ < 0.05, **$P$ < 0.01, ***$P$ < 0.005, $t$ test). All error bars indicate SEM. PE, phosphatidylethanolamine; PS, phosphatidylserine; PC, phosphatidylcholine; CL, cardiolipin.

between WT and MITOL KO brain, although the effect of MITOL KO on mitochondrial respiration in brain was no statistical significant difference (Fig S5A–E). It is known that enhanced reactive oxygen species (ROS) increases protein oxidation levels (Stadtman & Levine, 2003). We next examined whether MITOL controls ROS production using oxyblot analysis (Fig 4A). Unlike the activity of mitochondrial respiration, the level of oxidized proteins was clearly enhanced in the MITOL KO hippocampus. Although TUNEL-positive apoptotic cells were not detected in both WT and MITOL KO brain in the basal condition (date not shown), the expression of glial fibrillary acidic protein (GFAP) and Iba1, representing reactive astrocyte and activated microglia, consistently increased in the cerebral cortex and hippocampus of MITOL KO (Fig 4B–F). In particular, GFAP-positive area was increased in the MITOL KO cerebral cortex (Fig 4C). This induction of astrogliosis in MITOL KO brain was attenuated by treatment of a radical scavenger Edaravone (Fig 4G), suggesting that enhanced mitochondrial ROS in

MITOL KO brain leads to astrogliosis. A previous study has indicated that Cre recombinase activity is detected in astrocytes in several regions of the cerebral cortex of Emx1-Cre mice (Araya et al, 2008). We thus examined the effect of MITOL in primary astrocytes derived from WT or MITOL KO cerebral cortex. Indeed, MITOL protein was drastically reduced in the most of primary astrocytes, but not all, derived from MITOL KO brain (Fig S6A). MITOL KO primary astrocytes also exhibited increased small mitochondria (Fig S6B–E). Taken together, these findings indicate that MITOL deficiency in not only neurons but also astrocytes lead to gliosis due to enhanced oxidative stress.

## MITOL KO mice exhibited abnormal behavior

To mention the physiological role of MITOL, we investigated MITOL expression pattern in brain. Immunoblot analysis demonstrated a high expression of MITOL during brain development (Fig 5A).

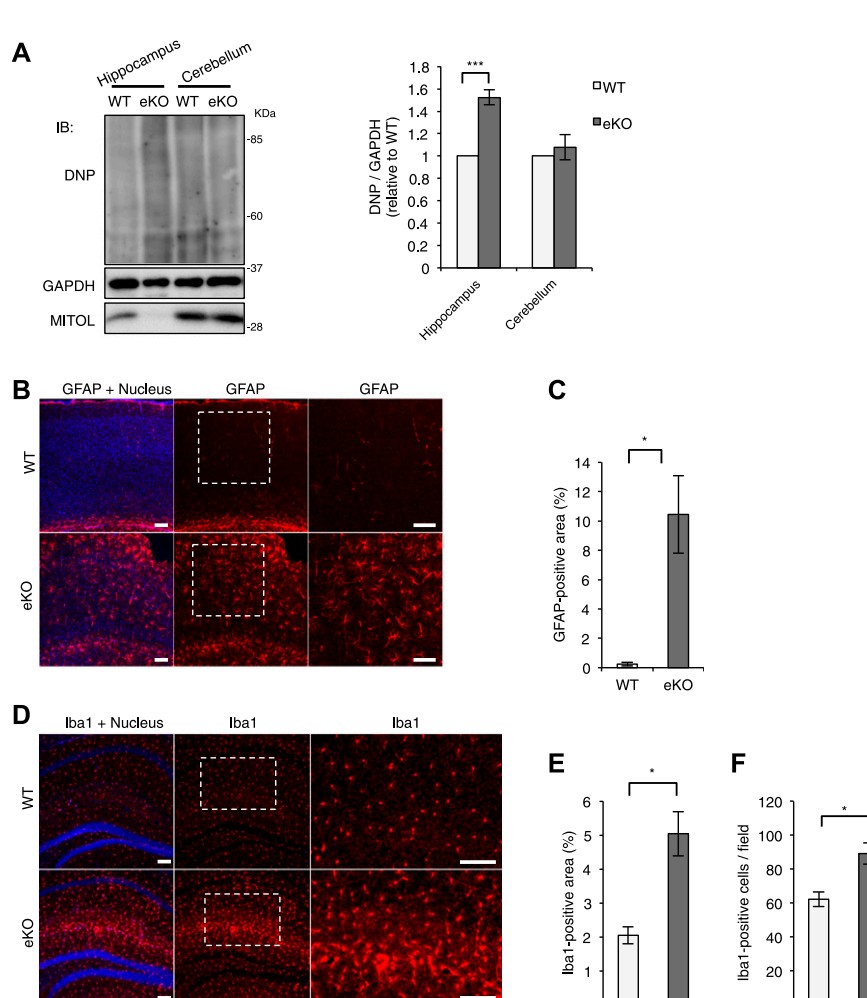

**Figure 4. Increased astrogliosis and microglial activation in MITOL eKO mice.**
**(A)** Accumulated carbonylated proteins in hippocampus but not in cerebullum of eKO. Whole cell extracts from hippocampus and cerebullum were subjected to oxyblot. Since carbonyl in the protein side chain were derivatized to 2,4-dinitrophenylhydrazone by reaction with 2,4-dinitrophenylhydrazine (DNPH), Oxidative protein levels were determined by immnoblotting using anti-DNP antibody. (n = 5; ***$P < 0.005$, $t$ test). Error bars indicate SEM. **(B)** Immunohistochemical staining of coronal sections across the cerebral cortical region from 8-wk-old mice using GFAP-specific antibodies. The right panels show high-magnification images of the boxed regions. Scale bars represent 100 $\mu$m. **(C)** The percentage of GFAP-positive area in (B) was quantified. (n = 4; *$P < 0.05$, $t$ test). Error bars indicate SEM. **(D–F)** Activation of microglia in eKO. **(D)** Immunohistochemical staining of coronal sections across the hippocampal region from 3-mo-old mice using Iba1-specific antibodies. The right panels show high-magnification images of the boxed regions. Scale bars represent 100 $\mu$m. **(E, F)** The percentage of Iba1-positive area (E) and the number of Iba1-positibe cells (F) were quantified. (n = 3 for WT and n = 4 for eKO; *$P < 0.05$, $t$ test). Error bars indicate SEM. **(G)** Radical scavenger rescued gliosis in eKO. 7-wks old mice were treatment with 2 mg/kg Edaravone for 5 d. Whole cell extracts from cerebral cortical of indicated mice were subjected to immunoblotting with indicated antibodies.

These data provide a possibility that MITOL plays an important role in brain development. To characterize eKO, we performed behavior tests. An open field behavioral test showed that eKO decreased the time spent grooming and increased immobile time at P21 (Fig 5B), suggesting a low ability to adapt to a new environment. No significant difference between WT and eKO was observed in jumping, distance moved and time spent in the center in the open field test. We next performed the elevated plus maze (EPM) test to measure anxiety using a plus-shaped maze with two open and two closed arms. There was no change in time spent in the arms between WT and eKO (Fig 5C). The Y-maze test showed reduced spontaneous alternation in eKO of both P21 and 3-mo-old males (Fig 5D), suggesting a poor short-term memory in eKO. Collectively, eKO exhibited mild behavioral changes, at least in part, due to neuroinflammation activated by abnormal mitochondrial morphology and impaired ER-mitochondria contact sites (Fig 5E).

## Discussion

### A link between mitochondrial dysfunction and developmental disorder

Mitochondrial dysfunction is associated with developmental disorders such as ASD (Frye & Rossignol, 2011; Rossignol & Frye, 2012). Several reports have demonstrated that mature neurons showed high-energy requirement to integrate their functional network, indicating that energy supply from mitochondria play the critical role in development and functions of neuron. However, a causal relationship between mitochondria and psychiatric disorders is largely unclear. Here, we showed for the first time that mitochondrial abnormalities in MITOL KO brain induced mild behavioral changes (Figs 1, 2, 3, 4, 5). MITOL KO brain also displayed enhanced oxidative stress, which may result from increased electron leakage from mitochondria (Fig 4). Mitochondrial abnormalities by MITOL KO appear to be more severe than a slight alternation of mitochondrial respiratory activity (Fig S5), suggesting that mitochondrial abnormalities by MITOL KO are responsible for ROS generation. However, there is a technical issue that mitochondrial fraction isolated from brain contains much mitochondria of glial cells and subtypes of neurons in which *mitol* gene is not deleted, therefore, the detailed investigation is required to understand the accurate role of MITOL in mitochondrial homeostasis in brain. We observed aberrant gliosis and microglial activation in MITOL KO brain, which are often detected in the brain of ASD patients (Sajdel-Sulkowska et al, 2011; Tetreault et al, 2012; Suzuki et al, 2013). Thus, pathological alterations in MITOL KO brain, such as gliosis, may partially cause behavioral changes. Importantly, aberrant gliosis in MTIOL KO brain was attenuated by treatment of a ROS scavenger Edaravone. Our results at least suggest that mitochondrial ROS, not only mitochondrial dysfunction in energy production, is sufficient to confer a potential risk for developmental disorder due to increased gliosis and microglial activation. Importantly, Emx1-Cre is expressed in astrocytes of some regions in the cerebral cortex (Araya et al, 2008).

Indeed, astrocytes in MITOL KO brain also exhibited the morphological abnormalities of mitochondrial network. Therefore, the morphological abnormalities and ROS generation of mitochondria in astrocytes, not only in neurons, might contribute to the excessive astrogliosis and developmental disorder.

### MITOL deficiency causes mitochondrial ROS generation via impairing mitochondrial network

In this study, mitochondrial abnormalities induced by MITOL KO is shown to trigger oxidative sterss-induced potential inflammatory state, indicating that MITOL KO confer a risk for inflammatory disease in brain. However, we cannot fully exclude the mechanism underlying enhanced oxidative stress in MITOL KO brain. In some reports using cell lines, increased production of mitochondrial ROS was observed when the *mitol* gene was specifically inhibited or deleted. Thus, oxidative stress in MITOL KO brain is considered to be resulted from incrased mitochondrial ROS. In both previous studies and this study, MITOL KO is thought to induce the accumulation of mitochondrial fission factor Drp1 and receptor Mid49. Interstingly, the expression pattern of MITOL correlates with that of Mid49, suggesting that MITOL tightly regulates mitochondrial size in developing brain throuhg Mid49. Thus, excessive mitochondrial fission was assumed as one cause underlysing increased oxidative stress in MITOL-KO brain. MITOL KO also lead to a significant reduction of ER-mitochondria contact sites due to Mfn2 inactivaiton. In yeast, the ER-mitochondrial encounter structure complex has been identified as tethering proteins between the ER and mitochondria and was found to be functionally important for phospholipid transports between the two organelles, thus, also required for CL biosynthesis in mitochondria (Kornmann et al, 2009). Indeed, ER-mitochondria contact sites function as a platform for lipid transport of phosphatidic acid, a CL precursor, from the ER to mtiochondria. A significant decrease in the level of CL by MITOL KO suggests that an efficient lipid transport via ER-mitochondria contact sites is required for the biosynthesis or metabolic turnover of CL species in brain. However, our data could not fully support the functional impairment of ER-mitochondria contact sites. Further study is needed to clarify the relationship between CL metabolism and ER-mitochondria contact sites in brain. Mitochondrial respiratory supercomplexes constitute an efficient energy producing system by decreasing electron leakage (Chen et al, 2008). CL is reported to be required for the formation of mitochondrial respiratory supercomplexes, although the role of CL in the formation of mitochondrial respiratory supercomplexes in vivo remain elucidated (Zhang et al, 2002; Bazan et al, 2013). Unexpectedly, BN-PAGE analysis confirmed that the formation of mitochondrial respiratory supercomplex structure was not drastically changed in MITOL KO brain. Because difference in molecular species of CL is observed between cell lines or tissues, the dominant type of CL species in mouse brain may be dispensable for the formation of mitochondriral respiratory supercomplexes. CL also maintains correct mitocondrial cristae morphology. Impaired cristae morphology in MITOL KO neurons, partly triggered by metabolic abnormalites of CL, may be another cause of aberrant oxidative stress.

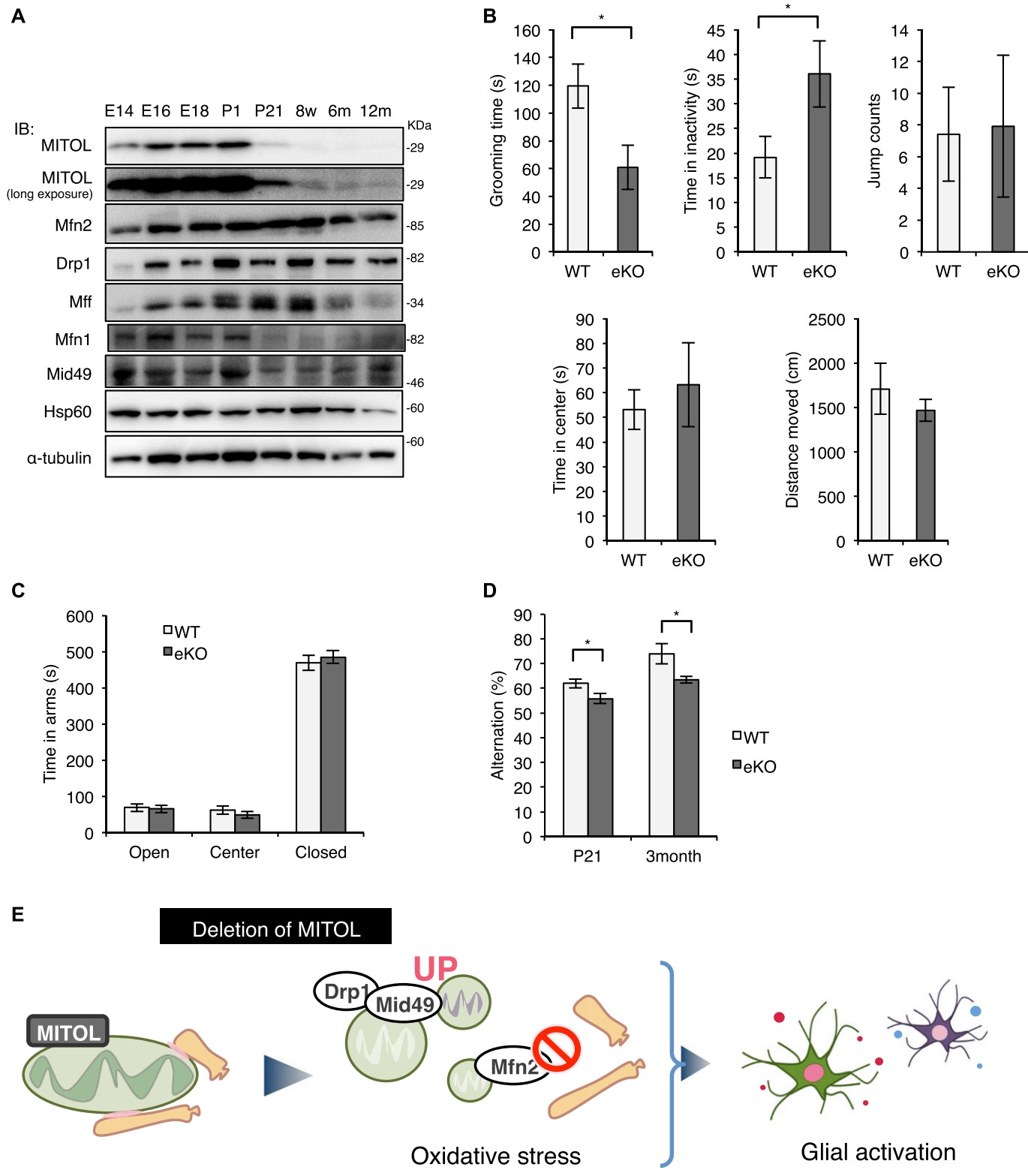

Figure 5. MITOL KO mice exhibit mild developmental disorder.
(A) High expression of MITOL in the developing neurons. Immunoblot analysis on whole cell lysates from the mouse cerebral cortex at each developmental stage with the indicated antibodies. (B) Reduced grooming time and increased immobility time in eKO at P21 in the open field test (n = 17 for WT and n = 12 for eKO; *$P < 0.05$, $t$ test). (C) No obvious change between WT and eKO in the elevated plus maze test. (n = 16 for WT and n = 17 for eKO at P21; *$P < 0.05$, $t$ test). (D) Reduced spontaneous alternation in eKO in the Y-maze test (n = 16 for WT and n = 17 for eKO at P21, n = 7 for WT and n = 5 for eKO at 3-mo old mice; *$P < 0.05$, $t$ test). All error bars indicate SEM. (E) Schematic model for the role of MITOL in vivo. MITOL induces the formation of ER-mitochondria contact sites by Mfn2 activation and promotes CL biosynthesis partly through

### ER-mitochondria contact sites determines mitochondrial size and morphological complexity

A serious concern was raised regarding the analytical and quantitative methods hitherto employed for analysis of the structures of ER-mitochondria contact sites. In previous studies, physical association between mitochondria and the ER or the sarcoplasmic reticulum, studied with transmission electron microscopy (TEM) (Raturi & Simmen, 2013), has been defined or measured to be distances of <15 nm (Filadi et al, 2015), 10–20 nm (Mannella, 2000), 14–20 nm (Naon et al, 2016), or <20 nm (Cosson et al, 2012); some measurement shows >30 nm in rat myocytes (Sharma et al, 2000). The merit of TEM is to analyze high-resolution of small area. In contrast, SBF-SEM can get a wide range of conformational information. In our study, the resolution of the images is 13 nm/pixel, and thus the direct contact in our study corresponds to close apposition of <13 nm, which is comparable to most of the studies about ER-mitochondria contact sites. Thus, our method using SBF-SEM can recognize the distance of ER tethering to mitochondria at almost the same level as the conventional studies. In addition, it has been found that 2D analyses of mitochondria, the ER, and ER-mitochondria contact sites inevitably lacked accuracy because these organelles actually assume complex 3D structures. Therefore, we use a 3D analysis for ER-mitochondria contact sites.

The first unexpected finding from the analyses including >500 mitochondria reconstructed from 300 slices at 40 nm axial intervals was the existence of branched, complex, and gigantic mitochondria in the cell body (Fig 1A–C). Many ER contact sites were observed for these branched, complex, and gigantic mitochondria. In some mitochondria, five or more contacts between the ER and mitochondria were identified per mitochondrion. Interestingly, the volume and number of these mitochondrial branchings were positively correlated with the number and total area of ER-mitochondria contact sites. This fact may suggest that ER-mitochondria contact sites are required to maintain the branched larged mitochondrion formed by mitochondrial fusion. Because membrane components, phospholipids, are required for the increase of mitochondrial volume, we proposed that the increase in lipid transport via ER-mitochondria contact sites also determined the volume of mitochondrion and intimately involved in mitochondrial maturation.

# Materials and Methods

### Mice

MITOL flox/flox mice were crossed with Emx1-Cre mice (Iwasato et al, 2004). Genotype was confirmed by tail tipping mice at around 1 mo. Mice were genotyped for *mitol* gene using PCR primer A, 5′-CAC AGG TAC GGT AGG TGT GTA AGC-3′, and primer B, 5′-ATG GGA ATG TGG TTC AGT TGT ACC-3′. 3-methyl-a-phenyl-2-pyrazolin-5-one (known as Edaravone), purchased from Sigma, was i.p. administered in

7-wk-old mice for 5 d (2 mg/kg per day). All animals were maintained under university guidelines for the care and use of animals. The experiments were performed after securing Tokyo University of Pharmacy and Life Sciences Animal Use Committee Protocol approval. Primary asctocytes were isolated from mice at postnatal day (P) 0–1 and maintained in DMEM supplemented with 10% FBS and penicillin/streptomycin at 37°C, in 5% $CO_2$, in a humidified chamber.as described previously (Kidana et al, 2018). To obtatin purified astrocytes, the glial culture was gently shaken for 6 h at 37°C.

### SBF-SEM analysis

SBF-SEM analyses of hippocampal neurons and corpus callosa of WT and eKO were performed as described previously (Ohno et al, 2014). Briefly, 12-wk-old mice were transcardially perfused with 4% PFA and 1% glutaraldehyde in 0.1 M PB (pH 7.4) under anesthesia, and incubated in the same fixative at 4°C over a few days. The fixed tissues were post-fixed and treated with osmium, thiocarbohydrazyde, uranyl acetate and lead aspartate, and embedded in resin. The specimens were imaged in field emission scanning electron microscopes, Merlin or Sigma (Carl Zeiss), equipped with 3View (Gatan). The obtained images were aligned and segmented with Fiji and TrakEM2 plugin (Cardona et al, 2012). Mitochondria and ER-mitochondria contact sites were reconstructed in 3D, and their distribution, area, branch, sphericity and volume were determined. ER-mitochondria contact sites is defined as direct attachments of mitochondrial outer membrane to continuous ER membrane (continuous in more than 10 slices). Reconstructed figures were prepared with Amira software (FEI Visualization Science Group).

### Immunofluorescence

Primary astrocytes were fixed with 4% PFA in PBS for 20 min at room temperature, then washed twice with PBS, permeabilized with 0.1% Triton X-100 in PBS for 15 min, then washed two times with PBS and blocked with 1% bovine serum albumin in PBS, all at room temperature. For double staining, the cells were incubated with indicated primary antibodies for 1 h at room temperature, washed three times with PBS, and then incubated with appropriate secondary antibodies for 30 min. The samples were washed as described above, mounted using Fluorescent Mounting Medium (Dako) and analyzed using an Olympus IX81 confocal fluorescence microscope. For mitochondrial area and number analysis, quantification was performed using the "Analyze particles" plugin in Fiji. Max projection images were processed with the "smooth" function in Fiji. The selected mitochondrial region of interest (ROI) (225 $\mu m^2$ ROI) was manually thresholded.

### Histology and immunohistochemistry

Brains were fixed in 4% PFA, and slice sections were prepared. Slices were incubated for two nights at 4°C with primary antibodies. Slices

ER-mitochondria contact sites-dependent efficient uptake of lipids from the ER to mitochondria. In addition, MITOL blocks excessive mitochondrial fission mediated by Mid49 and Drp1. Collectively, MITOL maintains mitochondrial network including ER tethering. These functions of MITOL in developing neurons prevents oxidative stress-induced glial activation, therefore, dysfunction of MITOL confers a risk for psychiatric disorders. E, Embrionic day; m, Month-old; P, Postnatal day; w, Week-old.

 Life Science Alliance

were incubated overnight with Alexa-conjugated secondary anti-bodies and counterstained with Hoechst. The samples were analyzed using an Olympus FU1000-D confocal fluorescence microscope.

## Open-field test

Locomotor activity was measured using an open-field test (40 × 40 × 25 cm). Mice were placed in the center of apparatus, and mouse movements were recorded with a video camera for 15 min: total distance traveled, grooming time, time in inactivity, time spent in the center (20 × 20 cm), and jumping behaviors. The arena was cleaned with 70% ethanol solution between the subjects.

## EPM test

The EPM consisted of two open arms (25 × 5 cm) and two enclosed arms of the same size, with 15 cm high transparent walls. The arms and central square were elevated to a height of 50 cm above the floor. Each mouse was placed in the central square of the maze (5 × 5 cm), facing one of the open arms. Mouse behaviour was recorded with a video camera for 10 min: time on open arm, entries into open arms.

## Y-maze test

The apparatus consists of three equal closed arms (each arm was 40 cm long, 14 cm high and 7 cm wide) converging to an equal angle. Each mouse was placed at the end of one arm and allowed to freely move through the maze during 10 min. An alternation was defined as consecutive entries in all three arms. The percentage of alter-nation was calculated.

## Antibodies

Anti-MITOL rabbit polyclonal antibodies was described previously (Yonashiro et al, 2006). Anti-$\alpha$-tubulin was from Sigma. Anti-VDAC1, anto-phospho-Drp1 (Ser616), and anti-succinate dehydrogenase complex subunit A (SDHA) antibodies were from Cell Signaling. Anti-Mfn2 and anti-calnexin (CNX) antibodies were from Santa Cruz Biotechnology. Anti-COX1 antibody was from Thermo Fisher Scientific. Anti-DNP antibody was from chemicon. Anti-ATP5a and anti-GAPDH antibodies were from Abcam. Anti-GFAP and anti-FundC1 antibody were from Millipore. Anti-Hsp60 antibody was from Enzo. Anti-Iba1 antibody was from Wako. Anti-Mid49, anti-Mff and anti-Tom20 antibodies were from Proteintech. Anti-NDUFA9 and anti-UQCRC2 antibodies were from Mitoscience. Anti-Drp1 antibody was from BD Bioscience. Anti-Mfn1 antibody was from BioLegend.

## Subcellular fractionation

Isolation of mitochondria was performed as described previously (Wieckowski et al, 2009).

## Western blotting

Cell lysates were separated by SDS-PAGE and transferred to the polyvinylidene difluoride membrane (Millipore). The blots were probed with the indicated antibodies, and protein bands on the blot were visualized by the enhanced chemiluminescence reagent (Millipore).

## BN-PAGE

100 $\mu$g Mitochondria isolated from cerebral cortex and hippo-campus of 8-wk-old mice or MEFs were solubilized with lysis buffer (0.25% n-Dodecyl b-D-Maltopyranoside [DDM] or 0.625% digitonin, 20 mM NaCl, 50 mM imidazole pH 7.0, 1 mM EDTA, 10% glycerol, 500 mM 6-aminocaproic acid, 0.005% Coomassie Brilliant blue G-250, protease inhibitors). After centrifugation at 8,000$g$ for 10 min at 4°C, the supernatants were electrophoresed through 3–10%, 3–12% or 4–15% polyacrylamide gradient gels. The gels were subjected to CBB staining or IB using the Abs described in the figures.

## Sucrose-gradient assay

Sucrose-gradient assay was performed as described previously (Sugiura et al, 2013).

## Lipid analysis

Lipids were extracted from isolated mitochondrial fraction from cerebral cortex and hippocampus of 8-wk-old mice according to the previous method (Aoki et al, 2004). In brief, 2 mg or 300 $\mu$g of mitochondria was suspended in 2 ml 0.1M KCl. After addition of 6 or 3 mL methanol, 3 or 1.5 mL chloroform and 30 $\mu$l 1% 2,6-Di-t-butyl-4-methylphenol, samples were incubated for 10 min at room temperature. After addition of 3 or 1.5 mL chloroform and 3 or 1.5 mL ultra pure water, samples were centrifuged for 5 min at 200 $g$. The chloroform (lower) phases were transferred into a glass tube. After addition of 3 or 1.5 mL chloroform to the upper phase of the sample, samples were centrifuged for 5 min at 200 $g$. The chloroform phase were combined to the glass tube. After the solvent was evaporated, lipids were resuspended in chloroform/methanol (2:1, vol/vol). For the determination of the mitochondrial lipid content, TLC was performed according to the previous method (Mourier et al, 2015). Lipids were separated using chloroform/methanol/glacial acetic acid 65:28:8 (vol/vol/vol) as a solvent system. For detection of lipid bands, the TLC plates were sprayed with a phosphoric acid/copper sulfate reagent (0.469 g of $CuSO_4(H_2O)_5$ and 1.765 mL of 14.6M $H_3PO_4$ in 10 mL of water) and charred at 160°C for 1 min. Lipid bands were quantified by ImageJ.

## Chromatographic and MS conditions

A Prominence series high-performance LC system (Shimadzu) containing a CBM-20A system controller, two LC-20AD pumps, a semi-micro gradient mixer, and a SIL-20AC autosampler was used. Separation was performed on an L-column2 ODS (3 $\mu$m, 2.1 mm i.d. × 100 mm) column (CERI). The column oven temperature was operated at 40°C. The mobile phase applied was the binary system of A (acetonitrile/trimethylamine/acetic acid, 100/0.5/0.5 vol%) and B (2-propanol/trimethylamine/acetic acid, 100/0.5/0.5 vol%) formulated as follows: 30% B to 95% B over 7.5 min and hold 95% B for 7.5 min at a flow rate of 200 $\mu$l/min. The injection volume of 5 $\mu$l was

used. A hybrid triple-quadrupole/linear ion trap mass spectrometer (3200 QTRAP LC-MS/MS, SCIEX, Concord) equipped with Turbo V ion source and Electrospray Ionization probe was used to analyze cardiolipin species in the samples. The source was operated in the negative ion mode with instrument parameters including curtain gas: 20, ion spray voltage: −4,500 V, declustering potential: −140 V, temperature: 500°C, nebulizer gas: 40, and heater gas: 30. The enhanced mass spectrum-information-dependent acquisition-enhanced product ion (EMS-IDA-EPI) operating mode was used to scanning metabolites. The EMS was obtained over a range from m/z 1,000 to 1,600. The enhanced product ion full-scan MS/MS spectra were used for confirmation of compound identifications via searching against authentic compounds and/or theoretical library of MS/MS spectra. Data analysis was performed with Analyst v1.5.2 software.

## Supplementary Information

## Acknowledgements

We thank Tetsuya Konishi and Yuki Yamamichi for technical assistance. This study was supported in part by MEXT/JSPS KAKENHI (to S Nagashima, R Inatome, T Fukuda, and S Yanagi) and MEXT-Supported Program for the Strategic Research Foundation at Private Universities (to S Nagashima, R Inatome and S Yanagi), JSPS KAKENHI, The Comprehensive Brain Science Network (CBSN), The Cooperative Study Programs of National Institute for Physiological Sciences (to N Ohno) and Research Grant from National Center of Neurology and Psychiatry (no. 30-5 to N Ohno), The Uehara Memorial Foundation, The Naito Foundation (to S Nagashima and S Yanagi), The Sumitomo Science Foundation, The Cosmetology Research Foundation, The Ono Medical Research Foundation, The Tokyo Biochemical Research Foundation and AMED under grants JP17gm5010002 and JP18gm5010002 (to S Yanagi).

## Author Contributions

N Shun: conceptualization, data curation, and writing—original draft.
T Keisuke: data curation and writing—original draft.
O Nobuhiko: data curation, formal analysis, and supervision.
I Satoshi: resources, data curation, and formal analysis.
A Motohide: data curation and formal analysis.
S Yurika: data curation and formal analysis.
T Takumi: data curation.
T Takeshi: data curation.
S Ayumu: validation.
F Toshifumi: validation.
M Nobuko: validation.
I Ryoko: data curation and writing—original draft, review, and editing.
Y Shigeru: conceptualization, supervision, funding acquisition, and writing—original draft, review, and editing

## Conflict of Interest Statement

The authors declare that they have no conflict of interest.

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
