## [Reviewer comments · Life Science Alliance]

Life Science Alliance

MITOL deletion in the brain impairs mitochondrial structure and ER tethering leading to oxidative stress

Shun Nagashima, Keisuke Takeda, Nobuhiko Ohno, Satoshi Ishido, Motohide Aoki, Yurika Saitoh, Takumi Takada, Takeshi Tokuyama, Ayumu Sugiura, Toshifumi Fukuda, Nobuko Matsushita, Ryoko Inatome, and Shigeru Yanagi

DOI: <https://doi.org/10.26508/lsa.201900308>

Corresponding author(s): *Shigeru Yanagi, Tokyo University of Pharmacy and Life Sciences*

Review Timeline:

Submission Date:	2019-01-16
Editorial Decision:	2019-02-27
Revision Received:	2019-06-07
Editorial Decision:	2019-07-09
Revision Received:	2019-07-31
Editorial Decision:	2019-07-31
Revision Received:	2019-08-01
Accepted:	2019-08-01

Scientific Editor: Andrea Leibfried

Transaction Report:

February 27, 2019

Re: Life Science Alliance manuscript #LSA-2019-00308-T

Prof. Shigeru Yanagi
Tokyo University of Pharmacy and Life Sciences
School of Life Sciences
1432-1 Horinouchi
Hachioji, Tokyo 192-0392
JAPAN

Dear Dr. Yanagi,

Thank you for submitting your manuscript entitled "MITOL deletion in brain disrupts MAM and mitochondrial structure leading to neuroinflammation" to Life Science Alliance. The manuscript was assessed by expert reviewers, whose comments are appended to this letter.

As you will see, the reviewers appreciate your data but think that some clarifications as well as a better analysis of mitochondria/mitochondrial function in neurons versus astrocytes/glial cells is needed (please see also comment from reviewer #2 and #3). We would thus like to invite you to provide a revised version of your work, addressing the reviewers comments. The requested staining for Drp1 may prove difficult on tissue sections and it is therefore not mandatory to address this request.

Thank you for this interesting contribution to Life Science Alliance. We are looking forward to receiving your revised manuscript.

Sincerely,

B. MANUSCRIPT ORGANIZATION AND FORMATTING:

Reviewer #1 (Comments to the Authors (Required)):

Nagashima and Yanagi studied MITOL function in neurons. They generated nerve-specific MITOL knockout mice for the first time and found disrupted MAM and loss of branched mitochondria

structures in MITOL-deficient neurons. The authors suggested that these morphological abnormalities of mitochondria in neurons increased the risk of developmental disorder through oxidative stress-induced neuroinflammation. Although the molecular mechanism is not clearly provided in this manuscript, the authors put enormous efforts to explore MITOL function in vivo and provide a novel and interesting aspect of MITOL function in neurons.

I only have one minor point.

Because MITOL interacts with mitochondrial fusion and fission proteins, I recommend you to change the words to "mitochondria dynamics" rather than "mitochondrial fission" in the abstract.
: A role of MITOL in both the formation of mitochondria-associated ER membrane (MAM) and mitochondrial fission prompted us----

Typos:

page 2: Edoplasmic reticulum --Endoplasmic reticulum

page 24: Refference Reference

Reviewer #2 (Comments to the Authors (Required)):

Nagashima et al

In this manuscript, Nagashima et al focus on the role of MITOL in the regulation of the contacts between ER and mitochondria, and its effect on mitochondrial respiratory complexes. Moreover, the authors show that MITOL ablation results in reduced cardiolipin concentration, increased ROS production and astrogliosis. Overall, the data shown in this work are certainly intriguing, although not compelling enough to support most of the author's conclusions:

- In fig. 1 the authors show that mitochondria are significantly fragmented in neurons from MITOL KO animals compared to WT. Their data also show increases in DRP1 on crude mitochondrial samples, which help them conclude that MITOL ablation might result into increases in DRP1-mediated mitochondrial fission. Given the cross-contamination of crude mitochondrial fractions with many other subcellular compartments, the authors should strengthen this conclusion by analyzing DRP1 localization on mitochondria using purified mitochondrial samples, or imaging approaches.
- In fig. 2., using three-dimensional reconstruction of negative-staining EM images on MITOL KO neurons, the authors show a significant reduction of ER-mito contacts. While this is a great approach to determine the degree of apposition between these organelles, it is insufficient to conclude that MAM, as a functional domain, is impaired. The authors should specify that their approach measured ER-mito contacts, but do not validate MAM functional disruption. To do that, it is necessary to assay one or more activities located at this domain.
- In fig. 3, please, report cardiolipin content as pmol/L or molarity percentage (molar fraction). Given the interconnection between lipid classes and species, Phospholipids/PC is not a valid measure.
- Mitochondrial function measurements are non-acceptable. Alteration in the levels of protein expression of respiratory complexes can be similar to WT even in the face of significant changes in oxygen consumption. Likewise, reporting in-gel activities is not an adequate approach to determine whether mitochondrial activity is affected. The authors should measure oxygen consumption to support their conclusion.

Anyhow, it is quite puzzling that in light of a significant cristae disruption and reductions in the concentration of cardiolipin, both of which are essential for the activity of respiratory complexes and supercomplexes, and increased ROS, the authors conclude that mitochondria function is not affected. To this regard, it is possible that mitochondrial respiration is only affected in neuronal populations compared to glia and microglia. Given the that ratio of glia/neuron is ~10, and the astrogliosis occurring in these mice, using homogenates of tissues will impede the authors to

unravel any potential underlying neuronal-specific mitochondrial alteration masked by the higher number of other proliferating brain cells.

Other,

- Some concepts in the introduction should be reviewed and corrected. For instance, "high demand of mitochondrial respiratory activity is accompanied by increased proton leak", is quite misleading.
- MAM's description as "the space between ER and mitochondria" should be fixed. MAM is the ER domain in close apposition to mitochondria, but it is still ER.
- Causal relationship between ROS and astrogliosis has not been shown. The authors should discuss this, but cannot conclude it with no specific data that links both processes.
- The language used in some sentences in the discussion are confusing and misleading.

Reviewer #3 (Comments to the Authors (Required)):

In a manuscript by Nagashima et al. the authors analyze consequences of depletion/knockout of mitochondrial E3 Ub ligase Mitol in cerebral cortex and hippocampus. They discovered an increase in oxidative stress in Mitol-deficient neurons/brains, which correlated with astrogliosis and microglial activation. Based on these data, the authors conclude that mitochondrial abnormalities in Mitol-deficient brains trigger neuroinflammation through oxidative stress. They also analyzed MAMs (mitochondria-associated ER membrane) and mitochondrial structure in vivo using three-dimensional EM microscopy (SBF-SEM microscopy). Since Mitol has been shown to control mitochondrial fission and MAM formation in cultured cells, most of the findings shown here are not really novel. Nevertheless, in vivo studies in brain-specific Mitol knockout presented in this report add some impact to this work. Thus, while this paper mostly confirms data obtained in other models, there are several strengths. The novel observations are behavioral changes, astrogliosis and microglial activation apparently resulting from Mitol deficiency. However, this is also the weakest part of the paper.

Overall data are of high quality and largely support the authors' conclusions. There is one major issue that should be addressed before this work can be recommended for publication (see below).

Specific comments:

1. The authors analyzed mitochondria morphology in hippocampal neurons. However, the data showing astrogliosis and microglial activation support the possibility that mitochondrial structure and function could be also affected in astrocytes and glia. This should be experimentally tested and discussed in the revised manuscript. Without these data the paper appears disconnected (e.g. investigation of mitochondrial structure and MAM formation in neurons, and functional studies focus on non-neuronal cells).
2. Perhaps term "psychiatric behavior", used in this manuscript in discussion of maze and other behavioral tests, could be replaced with "behavioral changes".

Reviewer #1 (Comments to the Authors (Required)):

Nagashima and Yanagi studied MITOL function in neurons. They generated nerve-specific MITOL knockout mice for the first time and found disrupted MAM and loss of branched mitochondria structures in MITOL-deficient neurons. The authors suggested that these morphological abnormalities of mitochondria in neurons increased the risk of developmental disorder through oxidative stress-induced neuroinflammation. Although the molecular mechanism is not clearly provided in this manuscript, the authors put enormous efforts to explore MITOL function in vivo and provide a novel and interesting aspect of MITOL function in neurons.

I only have one minor point.

Because MITOL interacts with mitochondrial fusion and fission proteins, I recommend you to change the words to "mitochondria dynamics" rather than "mitochondrial fission" in the abstract.

: A role of MITOL in both the formation of mitochondria-associated ER membrane (MAM) and mitochondrial fission prompted us----

Response:

We thank the positive comment of reviewer#1. Following the comment, the word "mitochondrial fission" in abstract was changed to the word "mitochondria dynamics" since MITOL regulates not only mitochondrial fission but also mitochondrial fusion as followed.

Abstract

"A role of MITOL in both the formation of mitochondria-associated ER membrane (MAM) and mitochondrial **dynamics** prompted us to characterize three-dimensional MAM structures *in vivo*."

Typos:

page 2: Edoplasmic reticulum --Endoplasmic reticulum

page 24: Refference Reference

Response:

We are very sorry for these careless mistakes. We corrected these errors.

Reviewer #2 (Comments to the Authors (Required)):

Nagashima et al

In this manuscript, Nagashima et al focus on the role of MITOL in the regulation of the contacts between ER and mitochondria, and its effect on mitochondrial respiratory complexes. Moreover, the authors show that MITOL ablation results in reduced cardiolipin concentration, increased ROS production and astrogliosis. Overall, the data shown in this work are certainly intriguing, although not compelling enough to support most of the author's conclusions:

- In fig. 1 the authors show that mitochondria are significantly fragmented in neurons from MITOL KO animals compared to WT. Their data also show increases in DRP1 on crude mitochondrial samples, which help them conclude that MITOL ablation might result into increases in DRP1-mediated mitochondrial fission. Given the cross-contamination of crude mitochondrial fractions with many other subcellular compartments, the authors should strengthen this conclusion by analyzing DRP1 localization on mitochondria using purified mitochondrial samples, or imaging approaches.

Response:

This opinion is right on target. However, it is technically too difficult to analyze the localization of DRP1 by imaging approaches using mouse brain. Also, as shown by many previous reports, DRP1 is rarely detected in the pure mitochondrial fraction isolated percoll-density centrifugation. It may be partly due to the DRP1 localization in the ER-mitochondria contact sites. We therefore consider that MITOL KO resulted in excessive mitochondrial fission by not only DRP1 but also its mitochondrial receptor Mid49 accumulated in the crude mitochondrial fraction of MITOL KO brain.

- In fig. 2., using three-dimensional reconstruction of negative-staining EM images on MITOL KO neurons, the authors show a significant reduction of ER-mito contacts. While this is a great approach to determine the degree of apposition between these organelles, it is insufficient to conclude that MAM, as a functional domain, is impaired. The authors should specify that their approach measured ER-mito contacts, but do not validate MAM functional disruption. To do that, it is necessary to assay one or more activities located at this domain.

Response:

Your concern is reasonable. Several reports have indicated that ER-mitochondria contact sites play central role in calcium transfer and lipid metabolism. Since it is difficult for us to perform Ca^{2+} imaging experiment using adult mice, we tried to monitor MAM function in the metabolism of cardiolipin (Figure 3), which is synthesized from phosphatidic acid supplied from the ER via ER-mitochondria contacts. We demonstrated the reduced cardiolipin species in MITOL-KO brain (Figure 3). Thus, we consider that MAM function involved in lipid metabolism was impaired in MITOL KO brain.

- In fig. 3, please, report cardiolipin content as pmol/L or molarity percentage (molar fraction). Given the interconnection between lipid classes and species, Phospholipids/PC is not a valid measure.

Response:

We collaborated with Dr. Aoki, an expert for lipid analysis. He judged that a qualitative data is enough to meet our purpose. To obtain quantitative data, it takes quite a lot of time. Please forgive us for this.

- Mitochondrial function measurements are non-acceptable. Alteration in the levels of protein expression of respiratory complexes can be similar to WT even in the face of significant changes in oxygen consumption. Likewise, reporting in-gel activities is not an adequate approach to determine whether mitochondrial activity is affected. The authors should measure oxygen consumption to support their conclusion. Anyhow, it is quite puzzling that in light of a significant cristae disruption and reductions in the concentration of cardiolipin, both of which are essential for the activity of respiratory complexes and supercomplexes, and increased ROS, the authors conclude that mitochondria function is not affected. To this regard, it is possible that mitochondrial respiration is only affected in neuronal populations compared to glia and microglia. Given the that ratio of glia/neuron is ~10, and the astrogliosis occurring in these mice, using homogenates of tissues will impede the authors to unravel any potential underlying neuronal-specific mitochondrial alteration masked by the higher number of other proliferating brain cells.

Response:

As the reviewer pointed out, whole brain contains many subtypes of neurons and glial cells including excitatory/inhibitory neurons, oligodendrocytes, astrocytes and microglia. Since astrogliosis and microglial activation were observed in MITOL KO brain, the mitochondrial fraction isolated from MITOL KO brain might contain much mitochondria of astrocytes and microglia than that from MITOL WT brain. We tried to solve this issue, but we found that the amount of neurons isolated from the brain of adult mouse is insufficient in quantity to examine the activity of mitochondrial respiratory complexes. We therefore conclude that it is technically difficult to address this issue at present. I hope you will understand.

Other,

- Some concepts in the introduction should be reviewed and corrected. For instance, "high demand of mitochondrial respiratory activity is accompanied by increased proton leak", is quite misleading.

Response:

We apologize that our explanation was confusing. According to your comments, we corrected some sentences in the introduction as follows.

Introduction

“high demand of mitochondrial respiratory activity is accompanied by a risk of oxidative stress due to increased electron leak from mitochondrial respiratory chain under physiological and pathological changes impaired mitochondrial homeostasis”

“high quality mitochondria are required for correct brain development and functions thereafter”

“Drp1 is an essential modulator of mitochondrial fission.”

- MAM's description as "the space between ER and mitochondria" should be fixed. MAM is the ER domain in close apposition to mitochondria, but it is still ER.

Response:

Thank you for your kind concern. We corrected these descriptions about MAM carefully as follows.

Introduction

“The unique membrane of the endoplasmic reticulum (ER) connected with mitochondria is known as the mitochondria-associated ER membrane (MAM)”

“Accumulating evidence suggest that the proximal junction between the ER and mitochondria plays multiple, important cellular functions not only in the efficient transfer of Ca²⁺ from the ER to the mitochondria and lipid metabolism but also the formation of the autophagic isolation membrane, cell death signaling and other processes through the MAM”

Result

“To identify MAM, the mitochondria and ER in each EM image were outlined in yellow and green, respectively, and ER membranes-attached mitochondria were labeled as MAM domains in red”

Discussion

“In yeast, the ERMES complex has been identified as tethering proteins between the ER and mitochondria...”

- Causal relationship between ROS and astrogliosis has not been shown. The authors should discuss this, but cannot conclude it with no specific data that links both processes.

Response:

We agree that this point is particularly important for our study. To investigate the causal relationship between ROS and astrogliosis, mice were treated with a ROS scavenger Edaravone. Importantly, excessive astrogliosis in MITOL KO brain was partly rescued by the treatment of Edaravone (Figure 4G). Thus, a likely conclusion could be that oxidative stress in brain lead to astrogliosis. We also mentioned it in the discussion part of our revised manuscript as followed.

Discussion

“Importantly, aberrant gliosis in MITOL KO brain was attenuated by the treatment of a ROS scavenger Edaravone.”

- The language used in some sentences in the discussion are confusing and misleading.

Response:

We are very sorry for confusing sentences. We corrected them carefully.

Reviewer #3 (Comments to the Authors (Required)):

In a manuscript by Nagashima et al. the authors analyze consequences of depletion/knockout of mitochondrial E3 Ub ligase Mitol in cerebral cortex and hippocampus. They discovered an increase in oxidative stress in Mitol-deficient neurons/brains, which correlated with astrogliosis and microglial activation. Based on these data, the authors conclude that mitochondrial abnormalities in Mitol-deficient brains trigger neuroinflammation through oxidative stress. They also analyzed MAMs (mitochondria-associated ER membrane) and mitochondrial structure in vivo using three-dimensional EM microscopy (SBF-SEM microscopy). Since Mitol has been shown to control mitochondrial fission and MAM formation in cultured cells, most of the findings shown here are not really novel. Nevertheless, in vivo studies in brain-specific Mitol knockout presented in this report add some impact to this work. Thus, while this paper mostly confirms data obtained in other models, there are several strengths. The novel observations are behavioral changes, astrogliosis and microglial activation apparently resulting from Mitol deficiency. However, this is also the weakest part of the paper. Overall data are of high quality and largely support the authors' conclusions. There is one major issue that should be addressed before this work can be recommended for publication (see below).

Specific comments:

1. The authors analyzed mitochondria morphology in hippocampal neurons. However, the data showing astrogliosis and microglial activation support the possibility that mitochondrial structure and function could be also affected in astrocytes and glia. This should be experimentally tested and discussed in the revised manuscript. Without these data the paper appears disconnected (e.g. investigation of mitochondrial structure and MAM formation in neurons, and functional studies focus on non-neuronal cells).

Response:

We are grateful for your insight that helped us strengthen our data. Since Emx1-Cre do not expressed in some subtypes of glial cells such as microglia and oligodendrocyte, we challenged to analyze the morphology of mitochondria in astrocytes. We consulted with Dr. Baba, who is an expert for astrocyte research, and found that it was very difficult to classify cells into neurons or astrocytes by SEM analysis. Thus, we examined the mitochondrial network in primary astrocytes. Consistent with previous reports, Emx1-Cre also induced the deletion of MITOL in the major of astrocytes (Supplemental

Figure 6A). In primary astrocytes isolated from MITOL KO brain, many mitochondria exhibited the spherical structure or showed the disconnected network overall (Supplemental Figure 6B-D). The morphological changes of mitochondria in astrocytes, not only in neurons, might contribute to the excessive astrogliosis in MITOL KO brain. We added the sentence described about the contribution of MITOL KO in astrocytes to astrogliosis in the discussion of revised manuscript. We are currently analyzing the contribution of MITOL KO in astrocytes to astrogliosis in detail and preparing it as a next project.

Discussion

Importantly, *Emx1-Cre* is expressed in astrocytes of some regions in the cerebral cortex (PMID: 18501628). Indeed, astrocytes in MITOL KO brain also exhibited the morphological abnormalities of mitochondrial network. Therefore, the morphological abnormalities and ROS generation of mitochondria in astrocytes, not only in neurons, might contribute to the excessive astrogliosis and developmental disorder.

2. Perhaps term "psychiatric behavior", used in this manuscript in discussion of maze and other behavioral tests, could be replaced with "behavioral changes".

Response:

We appreciate your suggestion. We corrected these sentences following your comment.

July 9, 2019

Re: Life Science Alliance manuscript #LSA-2019-00308-TR

Prof. Shigeru Yanagi
Tokyo University of Pharmacy and Life Sciences
School of Life Sciences
1432-1 Horinouchi
Hachioji, Tokyo 192-0392
Japan

Dear Dr. Yanagi,

Thank you for submitting your manuscript entitled "MITOL deletion in brain disrupts MAM and mitochondrial structure leading to neuroinflammation" to Life Science Alliance. The manuscript was assessed by expert reviewers, whose comments are appended to this letter.

As you will see, rev#1 and #3 appreciate the introduced changes, acknowledging that further insight would require extensive experimental revisions. Reviewer #2, however, is disappointed by the revision and has serious remaining concerns.

We have discussed your work in light of the remaining concerns and concluded that we can offer further consideration of your work here, pending additional satisfactory revision. Importantly, we think that the reviewer is correct in stating that you cannot refer to MAMs with the data at hand, even in light of your previous work => this needs to get changed to refer to "ER-mitochondria contact sites" throughout the manuscript text and figures. Furthermore, we agree that a qualitative comparison for the lipid analyses is needed, please re-normalize as requested by the reviewer. Finally, the other concerns still raised by the reviewer should get addressed by text changes. In addition to these revision points, please also add a callout in the manuscript text to Fig5E.

Thank you for this interesting contribution to Life Science Alliance. We are looking forward to receiving your revised manuscript.

Sincerely,

Andrea Leibfried, PhD
Executive Editor
Life Science Alliance
Meyerhofstr. 1
69117 Heidelberg, Germany

t +49 6221 8891 502
e a.leibfried@life-science-alliance.org
www.life-science-alliance.org

Reviewer #1 (Comments to the Authors (Required)):

The authors properly responded to the reviewers with additional experiments. Some of the experiments suggested by the reviewer cannot technically be addressed as the authors explained. I think that the revised manuscript is now ready to warrant publication.

Reviewer #2 (Comments to the Authors (Required)):

In this revised version of a manuscript, Nagashima et al focus on the role of MITOL in the regulation of the contacts between ER and mitochondria, and its effect on mitochondrial respiratory complexes. Moreover, the authors show that MITOL ablation results in reduced cardiolipin concentration, increased ROS production and astrogliosis.

I truly appreciate the efforts and the responses given by the authors to my comments. Alas, the data presented here are still not compelling enough to support the author's conclusions.

Previous comment: On fig. 1 the authors show that mitochondria are significantly fragmented in neurons from MITOL KO animals compared to WT. Their data also show increases in DRP1 on crude mitochondrial samples, which help them conclude that MITOL ablation might result into increases in DRP1-mediated mitochondrial fission. Given the cross-contamination of crude mitochondrial fractions with many other subcellular compartments, the authors should strengthen this conclusion by analyzing DRP1 localization on mitochondria using purified mitochondrial samples, or imaging approaches.

Answer from authors: DRP1 is rarely detected in the pure mitochondrial fraction isolated by percoll-density centrifugation. It may be partly due to the DRP1 localization in the ER-mitochondria contact sites. We therefore consider that MITOL KO resulted in excessive mitochondrial fission by not only DRP1 but also its mitochondrial receptor Mid49 accumulated in the crude mitochondrial fraction of MITOL KO brain.

Comment to authors' answer: Although these technical difficulties are understandable, the authors' conclusions should be supported by an additional read out, especially since Mid49 seems to be reduced after P21 in mice (figure 5A).

Previous comment: In fig. 2., using three-dimensional reconstruction of negative-staining EM images on MITOL KO neurons, the authors show a significant reduction of ER-mito contacts. While this is a great approach to determine the degree of apposition between these organelles, it is insufficient to conclude that MAM, as a functional domain, is impaired. The authors should specify that their approach measured ER-mito contacts, but do not validate MAM functional disruption. To do that, it is necessary to assay one or more activities located at this domain.

Answer from authors: Your concern is reasonable. Several reports have indicated that ER-mitochondria contact sites play central role in calcium transfer and lipid metabolism. Since it is difficult for us to perform Ca^{2+} imaging experiment using adult mice, we tried to monitor MAM function in the metabolism of cardiolipin (Figure 3), which is synthesized from phosphatidic acid supplied from the ER via ER-mitochondria contacts. We demonstrated the reduced cardiolipin species in MITOL-KO brain (Figure 3). Thus, we consider that MAM function involved in lipid metabolism was impaired in MITOL KO brain.

Comment to authors' answer: While I understand the technical difficulties, the data presented does not support the authors' conclusions. Cardiolipin levels are not a MAM functional read out. Without this, MAM dysfunction cannot be concluded from the data presented here.

Previous comment: In fig. 3, please, report cardiolipin content as pmol/L or molarity percentage (molar fraction). Given the interconnection between lipid classes and species, Phospholipids/PC is not a valid measure.

Answer from authors: We collaborated with Dr. Aoki, an expert for lipid analysis. He judged that a qualitative data is enough to meet our purpose. To obtain quantitative data, it takes quite a lot of time. Please forgive us for this.

Comment to authors' answer: I agree that quantitative data are not necessary, but I restate that the interconnection between lipid classes and species makes Phospholipids/PC not a valid measure. Please report CL levels normalized by a non-dependent factor. Alterations in CL should be reflected in the concentration(s) of other phospholipids such as PA or PS. As such, normalizing these data by total phospholipids is erroneous and misleading.

Previous comment: Mitochondrial function measurements are non-acceptable. Alteration in the levels of protein expression of respiratory complexes can be similar to WT even in the face of significant changes in oxygen consumption. Likewise, reporting in-gel activities is not an adequate approach to determine whether mitochondrial activity is affected. The authors should measure oxygen consumption to support their conclusion.

Anyhow, it is quite puzzling that in light of a significant cristae disruption and reductions in the concentration of cardiolipin, both of which are essential for the activity of respiratory complexes and supercomplexes, and increased ROS, the authors conclude that mitochondria function is not affected. To this regard, it is possible that mitochondrial respiration is only affected in neuronal populations compared to glia and microglia. Given the that ratio of glia/neuron is ~10, and the astrogliosis occurring in these mice, using homogenates of tissues will impede the authors to unravel any potential underlying neuronal-specific mitochondrial alteration masked by the higher

number of other proliferating brain cells.

Response: As the reviewer pointed out, whole brain contains many subtypes of neurons and glial cells including excitatory/inhibitory neurons, oligodendrocytes, astrocytes and microglia. Since astrogliosis and microglial activation were observed in MITOL KO brain, the mitochondrial fraction isolated from MITOL KO brain might contain much mitochondria of astrocytes and microglia than that from MITOL WT brain. We tried to solve this issue, but we found that the amount of neurons isolated from the brain of adult mouse is insufficient in quantity to examine the activity of mitochondrial respiratory complexes. We therefore conclude that it is technically difficult to address this issue at present. I hope you will understand.

Comment to authors' answer: This technical challenge is understandable. Then so, the text needs to be modified to reflect these shortcomings. It is well known that CL defects are highly correlated to OxPhos deficiencies and alterations in the supercomplex assembly. The authors' results are in opposition to a wide number of reports that have demonstrated this relationship in murine and human tissues. The authors need to address this significant problem.

Reviewer #3 (Comments to the Authors (Required)):

The authors addressed major issues. I do not believe that this work can be further significantly improved without extensive experimental work.

For Referee #2

In this revised version of a manuscript, Nagashima et al focus on the role of MITOL in the regulation of the contacts between ER and mitochondria, and its effect on mitochondrial respiratory complexes. Moreover, the authors show that MITOL ablation results in reduced cardiolipin concentration, increased ROS production and astrogliosis.

I truly appreciate the efforts and the responses given by the authors to my comments. Alas, the data presented here are still not compelling enough to support the author's conclusions.

<Point 1>

Previous comment: On fig. 1 the authors show that mitochondria are significantly fragmented in neurons from MITOL KO animals compared to WT. Their data also show increases in DRP1 on crude mitochondrial samples, which help them conclude that MITOL ablation might result into increases in DRP1-mediated mitochondrial fission. Given the cross-contamination of crude mitochondrial fractions with many other subcellular compartments, the authors should strengthen this conclusion by analyzing DRP1 localization on mitochondria using purified mitochondrial samples, or imaging approaches.

Answer from authors: DRP1 is rarely detected in the pure mitochondrial fraction isolated percoll-density centrifugation. It may be partly due to the DRP1 localization in the ER-mitochondria contact sites. We therefore consider that MITOL KO resulted in excessive mitochondrial fission by not only DRP1 but also its mitochondrial receptor Mid49 accumulated in the crude mitochondrial fraction of MITOL KO brain.

Comment to authors' answer: Although these technical difficulties are understandable, the authors' conclusions should be supported by an additional read out, especially since Mid49 seems to be reduced after P21 in mice (figure 5A).

Response:

According to this comment, we performed further experiments to obtain an additional data supporting mitochondrial accumulation of Drp1. It is widely accepted that a phosphorylation of Drp1 at Ser616 induces its mitochondrial recruitment and leads to mitochondrial fission. Since MITOL is considered to degrade Drp1 on mitochondria, we examined whether pDrp1-Ser616 is accumulated in MITOL KO brain. As expected, the level of pDrp1-Ser616 increased in MITOL KO brain (Figure S11). We believed that this data could overcome your concern partially.

<Point 2>

Previous comment: In fig. 2., using three-dimensional reconstruction of negative-staining EM images on MITOL KO neurons, the authors show a significant reduction of ER-mito contacts. While this is a great approach to determine the degree of apposition between these organelles, it is insufficient to conclude that MAM, as a functional domain, is impaired. The authors should specify that their approach measured ER-mito contacts, but do not validate MAM functional disruption. To do that, it is necessary to assay one or more activities located at this domain.

Answer from authors: Your concern is reasonable. Several reports have indicated that ER-mitochondria contact sites play central role in calcium transfer and lipid metabolism. Since it is difficult for us to perform Ca²⁺ imaging experiment using adult mice, we tried to monitor MAM function in the metabolism of cardiolipin (Figure 3), which is synthesized from phosphatidic acid supplied from the ER via ER-mitochondria contacts. We demonstrated the reduced cardiolipin species in MITOL-KO brain (Figure 3). Thus, we consider that MAM function involved in lipid metabolism was impaired in MITOL KO brain.

Comment to authors' answer: While I understand the technical difficulties, the data presented does not support the authors' conclusions. Cardiolipin levels are not a MAM functional read out. Without this, MAM dysfunction cannot be concluded from the data presented here.

Response:

Thank you for the comment. As you pointed out, a functional impairment of ER-mitochondrial contact sites in MITOL KO brain was not supported by current data in our manuscript. Thus, we added the sentence about the concern in the discussion section of the re-revised manuscript as below.

“However, our data could not fully support the functional impairment of ER-mitochondria contact sites. Further study is needed to clarify the relationship between CL metabolism and ER-mitochondria contact sites in brain.”

<Point 3>

Previous comment: In fig. 3, please, report cardiolipin content as pmol/L or molarity percentage (molar fraction). Given the interconnection between lipid classes and species, Phospholipids/PC is not a valid measure.

Answer from authors: We collaborated with Dr. Aoki, an expert for lipid analysis. He judged that a qualitative data is enough to meet our purpose. To obtain quantitative data, it takes quite a lot of time. Please forgive us for this.

Comment to authors' answer: I agree that quantitative data are not necessary, but I restate that the interconnection between lipid classes and species makes Phospholipids/PC not a valid measure. Please report CL levels normalized by a non-dependent factor. Alterations in CL should be reflected in the concentration(s) of other phospholipids such as PA or PS. As such, normalizing these data by total phospholipids is erroneous and misleading.

Response:

We appreciate your suggestion. We checked some reports analyzing phospholipids using TLC and found that our data normalized by PC caused misleading as you pointed out. Following previous reports analyzing phospholipids, the data were normalized by a different method as shown in Fig. 3D, Fig. S4A. We applied equal amount of lipid samples lysed from equal protein amount of mitochondria fraction and determined the quantity of each phospholipid spot without any normalization (Fig. 3D). Additionally, the

quantitative data normalized by total lipid was shown in Fig. S4A. Similar levels of total signals of phospholipids and non-separated spot (front spot) were observed between WT and MITOL KO brain (Fig. 3D). CL signal in MITOL KO brain was specifically reduced among all signals of phospholipids compared to those of WT brain (Fig. 3D). A similar result was obtained even when phospholipid signals were normalized by total lipid (Fig. S4A). Therefore, we concluded that MITOL KO in brain resulted in the reduction of CL.

<Point 4>

Previous comment: Mitochondrial function measurements are non-acceptable. Alteration in the levels of protein expression of respiratory complexes can be similar to WT even in the face of significant changes in oxygen consumption. Likewise, reporting in-gel activities is not an adequate approach to determine whether mitochondrial activity is affected. The authors should measure oxygen consumption to support their conclusion. Anyhow, it is quite puzzling that in light of a significant cristae disruption and reductions in the concentration of cardiolipin, both of which are essential for the activity of respiratory complexes and supercomplexes, and increased ROS, the authors conclude that mitochondria function is not affected. To this regard, it is possible that mitochondrial respiration is only affected in neuronal populations compared to glia and microglia. Given the that ratio of glia/neuron is ~10, and the astrogliosis occurring in these mice, using homogenates of tissues will impede the authors to unravel any potential underlying neuronal-specific mitochondrial alteration masked by the higher number of other proliferating brain cells.

Response: As the reviewer pointed out, whole brain contains many subtypes of neurons and glial cells including excitatory/inhibitory neurons, oligodendrocytes, astrocytes and microglia. Since astrogliosis and microglial activation were observed in MITOL KO brain, the mitochondrial fraction isolated from MITOL KO brain might contain much mitochondria of astrocytes and microglia than that from MITOL WT brain. We tried to solve this issue, but we found that the amount of neurons isolated from the brain of adult mouse is insufficient in quantity to examine the activity of mitochondrial respiratory complexes. We therefore

conclude that it is technically difficult to address this issue at present. I hope you will understand.

Comment to authors' answer: This technical challenge is understandable. Then so, the text needs to be modified to reflect these shortcomings. It is well known that CL defects are highly correlated to OxPhos deficiencies and alterations in the supercomplex assembly. The authors' results are in opposition to a wide number of reports that have demonstrated this relationship in murine and human tissues. The authors need to address this significant problem.

Response:

As you pointed out, accumulating evidence suggests a link between reduced CL and OxPhos dysfunction. Likewise, several reports have suggested that the amount of CL correlates to OxPhos activities as mentioned in your comment. Importantly, MITOL KO showed an approximately 30 % reduction of CL in brain. We consider that the level of CL reduction in MITOL KO brain is very mild compared to that in other knockout/transgenic mice showing both CL and OxPhos defects. In addition, mitochondrial fraction isolated from brain contains mitochondria derived from many subtypes of neurons and glial cells, although some subtypes express MITOL protein normally. Due to these reasons, MITOL KO brain may reveal slight decreases of respiratory complexes and activities with statistically non-significant difference. We added these concerns in the discussion section of re-revised manuscript following your suggestion as below.

“However, there is a technical issue that mitochondrial fraction isolated from brain contains much mitochondria of glial cells and subtypes of neurons in which MITOL gene is not deleted, therefore, the detailed investigation is required to understand the accurate role of MITOL in mitochondrial homeostasis in brain.”

July 31, 2019

RE: Life Science Alliance Manuscript #LSA-2019-00308-TRR

Prof. Shigeru Yanagi
Tokyo University of Pharmacy and Life Sciences
School of Life Sciences
1432-1 Horinouchi
Hachioji, Tokyo 192-0392
Japan

Dear Dr. Yanagi,

Thank you for submitting your revised manuscript entitled "MITOL deletion in brain impairs mitochondrial structure and ER tethering leading to oxidative stress". I now assessed the changes introduced in revision. I think that ideally, a normalization of the lipids to PS concentrations would have better addressed the concern of reviewer #2. The qualitative data and the mild effects on the level of PS/PE/PC phospholipid species as shown in Fig S4A are however convincing enough to move forward with your paper now. I also appreciate the text changes introduced and the added p-Drp1 western blot, and I would thus be happy to publish your paper in Life Science Alliance.

Please log in one more time to fill in the electronic license to publish form. Your manuscript number will change to LSA-2019-00308-TRRR, please make sure to move all manuscript files to this new version.

A. FINAL FILES:

B. MANUSCRIPT ORGANIZATION AND FORMATTING:

Sincerely,

August 1, 2019

RE: Life Science Alliance Manuscript #LSA-2019-00308-TRRR

Prof. Shigeru Yanagi
Tokyo University of Pharmacy and Life Sciences
School of Life Sciences
1432-1 Horinouchi
Hachioji, Tokyo 192-0392
Japan

Dear Dr. Yanagi,

Thank you for submitting your Research Article entitled "MITOL deletion in brain impairs mitochondrial structure and ER tethering leading to oxidative stress". It is a pleasure to let you know that your manuscript is now accepted for publication in Life Science Alliance. Congratulations on this interesting work.

*****IMPORTANT:** If you will be unreachable at any time, please provide us with the email address of an alternate author. Failure to respond to routine queries may lead to unavoidable delays in publication.*******

DISTRIBUTION OF MATERIALS:

Again, congratulations on a very nice paper. I hope you found the review process to be constructive and are pleased with how the manuscript was handled editorially. We look forward to future exciting submissions from your lab.

Sincerely,

Andrea Leibfried, PhD
Executive Editor
Life Science Alliance
Meyerohofstr. 1
69117 Heidelberg, Germany
t +49 6221 8891 502
e a.leibfried@life-science-alliance.org
www.life-science-alliance.org